# Flow interactions lead to self-organized flight formations disrupted by self-amplifying waves

Joel W. Newbolt[1], Nickolas Lewis[1], Mathilde Bleu[1], Jiajie Wu[1], Christiana Mavroyiakoumou [1], Sophie Ramananarivo [2] & Leif Ristroph [1] ✉

Collectively locomoting animals are often viewed as analogous to states of matter in that group-level phenomena emerge from individual-level interactions. Applying this framework to fish schools and bird flocks must account for visco-inertial flows as mediators of the physical interactions. Motivated by linear flight formations, here we show that pairwise flow interactions tend to promote crystalline or lattice-like arrangements, but such order is disrupted by unstably growing positional waves. Using robotic experiments on "mock flocks" of flapping wings in forward flight, we find that followers tend to lock into position behind a leader, but larger groups display flow-induced oscillatory modes – "flonons" – that grow in amplitude down the group and cause collisions. Force measurements and applied perturbations inform a wake interaction model that explains the self-ordering as mediated by spring-like forces and the self-amplification of disturbances as a resonance cascade. We further show that larger groups may be stabilized by introducing variability among individuals, which induces positional disorder while suppressing flonon amplification. These results derive from generic features including locomotor-flow phasing and nonreciprocal interactions with memory, and hence these phenomena may arise more generally in macroscale, flow-mediated collectives.

The collective movements of organisms inspire investigations into how novel behaviors of ensembles emerge from interactions among their members[1–9]. Bird flocks and fish schools involve interactions that are social or behavioral as well as physical interactions that are mediated by the aero- or hydro-dynamics of locomotion[4,10]. The challenges of understanding flow interactions stem from the spatiotemporal complexities of unsteady locomotion at intermediate to high Reynolds numbers Re[11,12], many of which are compounded when multiple bodies interact through their flow fields[13–16].

Understanding collective phenomena can benefit from devising physical analogues that abstract some elements from biological systems while potentially offering advantages in terms of control,

characterization and observation. This strategy has been successfully applied to systems with microscopic constituents such as swarms of swimming bacteria, which display turbulence-like motions that have been reproduced in suspensions of active microparticles and in related models and computational simulations[17–19]. Abiotic realizations, while idealizations, can help to identify and understand generic phenomena that are relatively insensitive to system details. Such knowledge can inform whether collectively moving ensembles may be usefully viewed as forms of active matter, a framework that promotes an exchange of ideas and methods with statistical mechanics and material physics[20–22].

Mechanical analogues can be directly informative on the fluid dynamical interactions involved in schools and flocks, and they can

[1]New York University, Courant Institute, Applied Math Lab, New York, USA. [2]LadHyX, Ecole Polytechnique, Institut Polytechnique de Paris, Paris, France. ✉e-mail: ristroph@cims.nyu.edu

also help to interpret behavioral aspects by establishing what flow effects must be contended with and what might be exploited. Robotic systems can serve these roles[23] and especially so-called robophysical experiments whose aim is not autonomous motion but rather to inform on physical mechanisms[24]. Research aimed at a single loco-motor has found that mechanized fins, wings, or foils driven to flap share with biological propulsors many aspects of the flows, forces, energetics, and dynamics[12,25,26]. Further, recent experimental, computational, and modeling work focusing on multi-propulsor interactions make this system a promising candidate for the study of high-Re collective locomotion[27–38].

Here, we use "mock flocks" or groups of self-propelling foils to investigate the structural and dynamical consequences of interactions through visco-inertial flows. We focus on linear formations in which many bodies are arranged serially or in tandem and whose dynamics within the formation are free and interactive[23,28,31,34,36,39]. Among all configurations, this setting is perhaps the most amenable to a detailed treatment of the flow-structure interactions, since the formation and dynamical freedom are one-dimensional (1D) and the flows quasi-2D. This system also draws inspiration from the so-called columnar formations adopted by many species of birds[40–42], which are well documented to occur but whose detailed structure and dynamics do not seem to have been studied. Some motivating photographs are shown in Fig. 1. Focusing on strictly in-line formations is an idealization that leverages previous investigations into tandem locomotion of pairs of foils, for which a follower has been shown to lock into specific positions behind a leader[27,29,30,32,33]. The group-wide consequences of collective flow interactions within long chains of flapping flyers are the subject of this investigation.

## Results

### Crystalline structures and collective excitations in robotic flocks

The idealized flock problem pertains to an in-line array of flapping flyers that self-propel and interact through their collective flow field. As illustrated in Fig. 2a, b, we consider foils driven with prescribed vertical oscillations (heaving-and-plunging motions) that interact

through the fluid and dynamically select their horizontal motions. Our experiments involve the related rotational setup shown in Fig. 2c, which extends previous flapping-foil systems[28,29,33,35,43,44] to accommodate many bodies, here up to five. The foils fan out radially from a common vertical shaft that is driven to flap up and down by a motor, this motion imparted to the foils. Each foil mounts to the shaft via an independent set of rotary bearings, permitting free rotation under the fluid forces on the foil. Flight thus takes the form of orbits around a water tank, these motions recorded on video that is analyzed to yield the positions of all members through time. Supplementary Video 1 introduces the key aspects of the experimental system. Additional information about the apparatus and procedures are provided as "Methods", including a review of previous studies that have validated rotational or "flight mill" systems against rectilinear locomotion[27–29,33,34,43–46]. The conditions studied here lead to motions of Reynolds number Re ~ $10^3$, which is on the lower end of the range displayed by schooling fish and flocking birds[11,25,47].

When driven, the foils take off in flapping flight around the tank, their front-back asymmetry dictating the direction. The collective motions can be sensitive to the initial conditions. It is often observed that one foil (typically the last) catches up with and crashes into its upstream neighbor. The two thereafter remain together and form an effectively larger wing that propels faster than the others and thereby aggregate all members through rear-end collisions. However, carefully spacing the foils and gradually ramping up the flapping frequency can lead to groups that last for tens of minutes or 1000s of flapping cycles. Supplementary Video 2 is a top-view recording of such a long-lived flock, and Video 3 captures the later collisions that cause the collapse of the group. These observations suggest that formation locomotion represents a metastable state of the system that can persist for long times but eventually succumbs to disturbances.

Measurements of the group structure confirm that the formation is remarkably well ordered. Representative time-series data for the positions $X_n(t)$ of the members of rank $n \in [1, 2, . . . , N]$ in a group of size $N = 5$ are shown in Fig. 2d. Here the lattice-like formation manifests as the nearly uniform distances separating successive members.

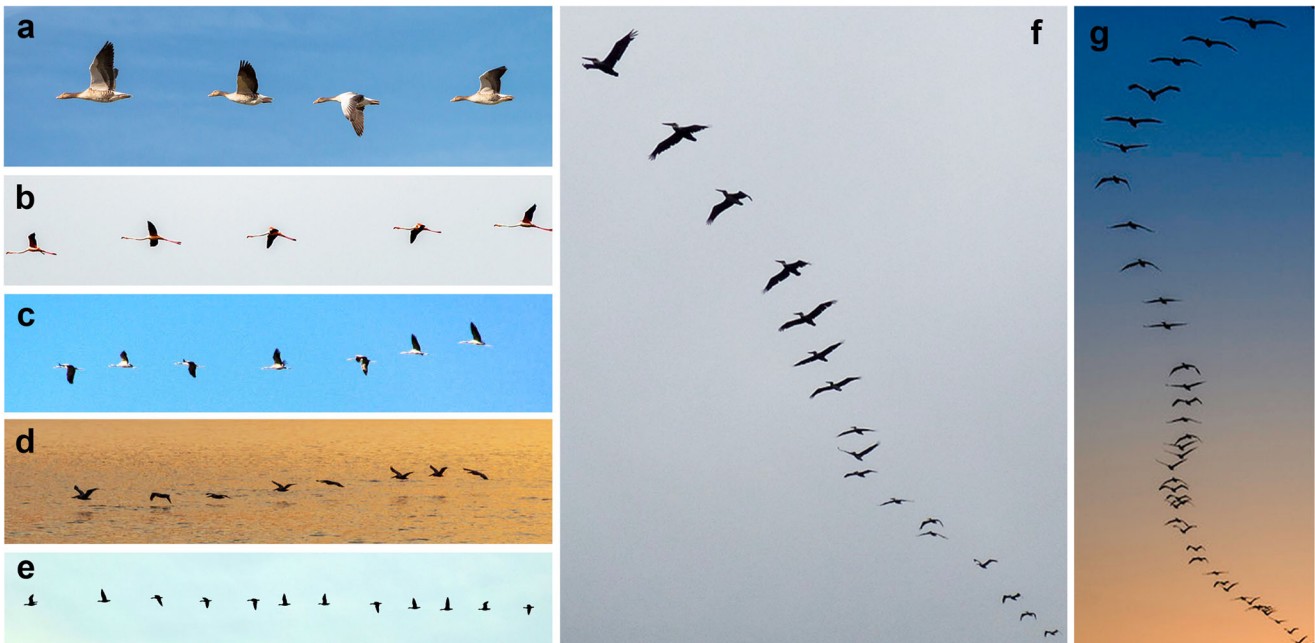

**Fig. 1 | Motivating examples of quasi-linear flight formations of birds.** Formations with varying numbers of members occur across several species, shown here in photographs of geese (**a** and **e**), flamingos (**b**), cranes (**c**), and pelicans (**d**, **f** and **g**). Directly in line and staggered or echelon arrangements are observed, and there seems to be no strict positional ordering along the flock nor phase relationship in the wing flapping motions. Photo credits to, respectively: E. Fleck, D. McGillicutty, H. Hajihusseini, M. South, V. Grigorev, G. Hayes, and G. Woolnough.

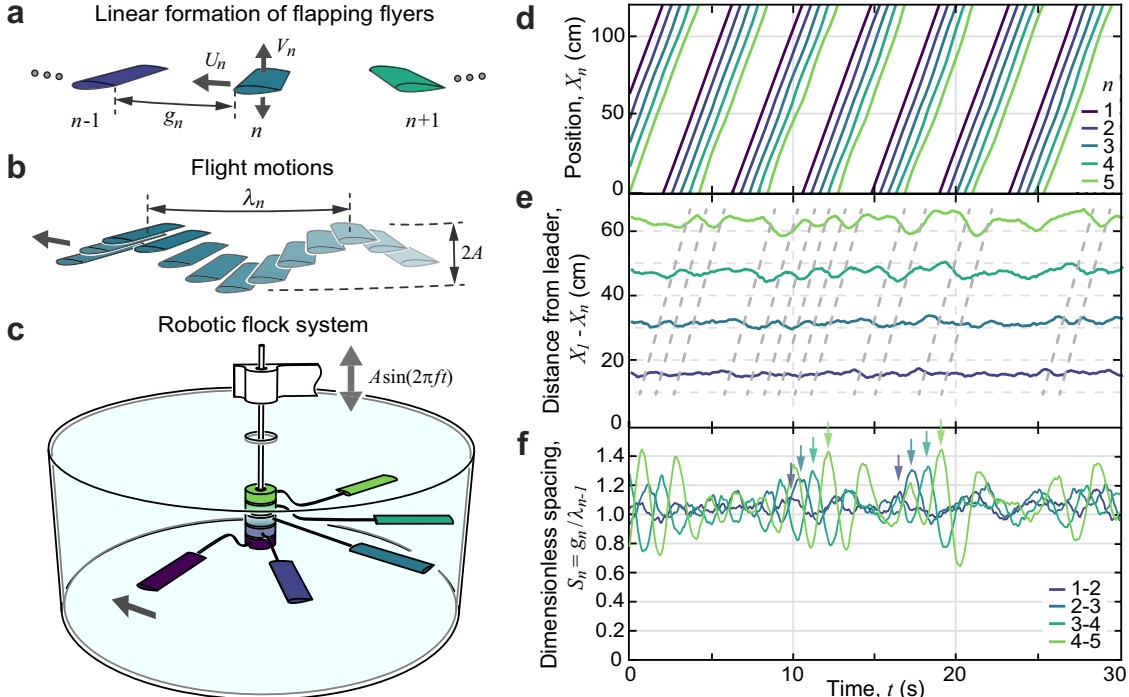

**Fig. 2 | Linear flight formations and a robophysical experiment. a** Idealized problem of a linear formation of flapping foils or wings indexed by $n$ and each with prescribed vertical oscillations and free forward flight motions. **b** Flapping leads to an undulatory trajectory of wavelength $\lambda$. **c** Experimental apparatus. Sinusoidal up-and-down oscillations of an upright shaft are driven by a motor system (not shown), thereby imparting flapping motions to multiple foils that fan out around a water tank. Each foil connects to the shaft via a support arm and low-friction rotary bearings, permitting free propulsion around the tank. **d** Measured motions of five foils during repeated orbits, where angular locations are converted to arc-length positions $X_n(t)$. **e** Positions measured relative to the leader $n = 1$ reveal stronger fluctuations for downstream members. Dashed lines highlight disturbances that propagate and amplify down the group. **f** The dimensionless inter-member gap spacing $S$ reveals that the foils tend to self-organize into a crystal-like formation with lattice parameter $\lambda$, but the members oscillate strongly during intermittent bursts of activity (arrows).

Supplementary Video 4 is a reanimation of these same data. The group structure is made more apparent in the plot of Fig. 2e displaying the positions relative to the leader ($n = 1$). Such spontaneous ordering into crystalline arrangements is consistent with previous experiments on foil pairs[29,33] and simulations of tandem arrays[27,30–32], all of which support the so-called Lighthill conjecture that locomotion formations may be induced or assisted by flow interactions[48].

The measurements of Fig. 2e also reveal relative motions between the foils that give important clues to the fragility of the group. Spacing fluctuations are seen to be larger in amplitude for downstream members, occasionally bringing the last two members into close proximity. Moreover, the fluctuations are correlated in time and across members. In Fig. 2e we highlight with dashed lines several dynamical events consisting of changes in relative position and which impact members at times that are delayed in proportion to their rank. Hence, disturbances propagate down the group much like traveling waves.

The dynamical correlations are made more apparent by tracking the gap spacing $g_n = X_{n-1} - X_n - c$ between successive pairs of foils (chord length $c$) with rank $n$ and $n-1$, as defined schematically in Fig. 2a. Figure 2f displays the gap dynamics as normalized by the wavelength $\lambda_{n-1}$ of the undulatory trajectory of the leading member in each pair (see Fig. 2b). The data show that the resulting dimensionless spacing $S_n = g_n/\lambda_{n-1}$ has time-averaged value of about 1 for all members. Hence the trajectory wavelength serves as the appropriate lattice parameter for the crystalline formation, consistent with previous findings for tandem pairs[29,33]. Most importantly, the fluctuations are temporally structured, showing up as oscillations of $S_n$ that occur in sequence and are more pronounced for members further downstream. The arrows of Fig. 2f highlight these features for representative events, which show up as transient increases in the spacing fluctuations. The dynamics within these events have the hallmarks of displacement waves of the traveling, longitudinal type. Namely, the member-to-member gap distances oscillate in time, the oscillations are delayed between members in a way that indicates downstream propagation, and the oscillations are along the propagation direction. Unlike conventional traveling waves, the amplitude of the excursions grows during transit.

Such dynamics were not reported in previous experiments on multiple foils rigidly linked to one another[28], because independent locomotion is essential. These dynamics were also not observed for pairs of independently propelling foils[29,33], presumably because sufficiently many members are needed to observe large-amplitude excitations and their wave-like mode of transmission. These dynamics were not reported in recent simulations at lower Re = 200 of many in-line flappers moving into quiescent fluid[31], suggesting that higher Reynolds numbers and/or ambient disturbances may be important.

As collective excitations that propagate on a lattice, these dynamics share some general features with conventional longitudinal displacement waves, such as sound waves or phonons in atomic and molecular crystals and related systems[49,50]. However, the distinctive properties of the cooperative motions seen here, especially their one-way transmission and amplification, derive from high-Re flow interactions and the associated non-potential forces. We introduce the term *flonons* for such flow-mediated fluctuations among flapping, flocking flyers. Their causes, characteristics, and consequences are investigated in what follows.

## Probing flow interactions by applying DC and AC perturbations

The robophysical system provides means for applying controlled and targeted perturbations to the flock towards understanding the origin

of the emergent formations and their dynamics. We first consider the application of a steady force to one member during flight[29], which is accomplished with the apparatus illustrated in Fig. 3a. Targeting member 2, the foil is first locked to the shaft via a set screw in its bearing housing but still propels freely thanks to another set of bearings that connects the upper portion of the shaft to the frame (white structure above the tank in Fig. 3a) that is oscillated by the motor. A mass-string-pulley system is then used to impart a torque whose magnitude is controlled by the hanging weight and whose direction is set by the sense in which the string is wound around a spool on the shaft. Complete details are given as "Methods". If sufficiently small, the external load causes the foil to take up a new position relative to its neighbors, where it resumes steady flight as the applied torque comes into balance with a hydrodynamic torque that reflects the flow interactions. Converting to effective forces at the foil mid-span, the force-displacement curve is determined from repeated measurements of the modified position for differing magnitudes of the load applied in both directions. In this way, the interaction force landscape is mapped out.

The data sets of Fig. 3c correspond to the latter 3 members in a group of 4, and each force-displacement curve reveals a spring-like response that stabilizes the position with $S \approx 1$. Applied forces $F$ that drive a given foil towards its upstream neighbor (smaller $S$) are resisted by increasingly strong hydrodynamic forces of the same magnitude, until a maximum beyond which the foil is driven into a collision. Similarly, force applied downstream leads to repositioning to greater $S$ up to the point in which $F$ reaches a minimum beyond which the foil

breaks free of its upstream neighbor. Strikingly, this response differs little across all members not in the leading position, as indicated by the high degree of overlap among the three data sets in Fig. 3c. Discernible differences appear only for the largest strains, for which the force is somewhat weaker on foils with multiple members upstream. Nonetheless, the general correspondence implies that the flow interactions are largely insensitive to the number of upstream and downstream neighbors. The force-displacement profile is also in good agreement with previous measurements of the force on the follower in a pair[29].

These results are indicative of nearest-neighbor, nonreciprocal hydrodynamic interactions. That is, the ambient flows that perturb any member $n$ can be attributed predominantly to its immediate upstream neighbor $n-1$. The nonreciprocity is well approximated by one-way interaction from leader to follower. We also note that the nearest-neighbor property applies to the direct interactions mediated by flows, whereas more distant members affect one another indirectly through the dynamics of intermediates. Hence, the group-wide consequences are not immediately apparent from knowledge of the member-to-member flow interactions.

Important additional insights can be gained by applying oscillatory perturbations to the leader during flight. As shown in Fig. 3b, such AC perturbations are accomplished by now locking foil 1 to the shaft, whose top end is fitted with a motor-driven oscillator that induces back-and-forth oscillations of the leader during its flight (see "Methods"). The resulting positional fluctuations of downstream members are measured. These data are characterized in Fig. 3d through the

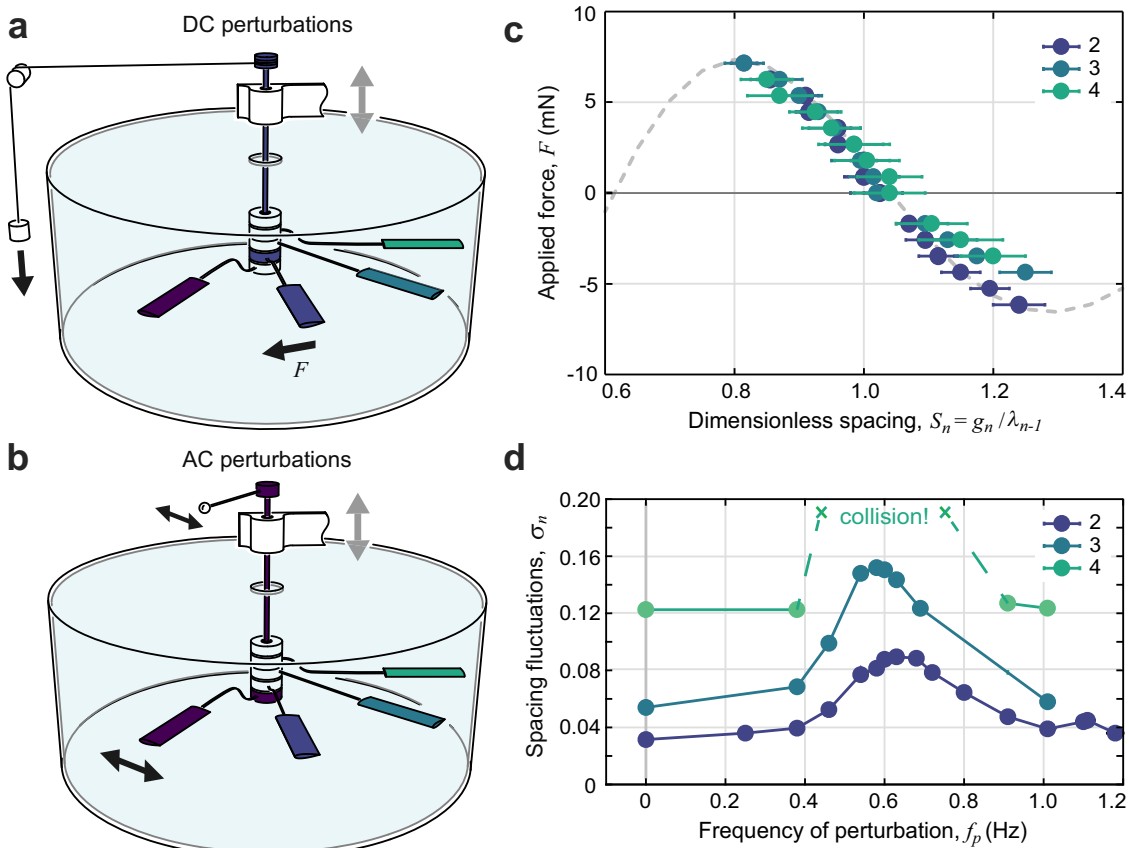

**Fig. 3 | Perturbing flocks to infer interactions and determine their consequences.** **a** Experimental means for applying a steady force to one member during flight[29]. The scheme is illustrated for the case of foil 2, who is first locked to the vertical shaft via a set screw in its bearing housing. A mass-string-pulley assembly provides an external load, whose magnitude is controlled by the hung weight and direction by the sense of winding of the string around the spool. **b** Means for applying oscillatory forces to the leader. The leader is locked to the shaft, which is fitted with a battery and motor that oscillate a mass. **c** Measured force-displacement profiles for members 2, 3, and 4 in a group of 4. The dashed curve is a guide to the eye. **d** Response of downstream members to an AC perturbation, as quantified by the standard deviation of the dimensionless spacing versus the perturbation frequency. The curves pertain to the last member in groups of size 2, 3, and 4. Later members resonate with amplitudes that increase with rank.

standard deviation $\sigma_n$ of the dimensionless spacing $S_n$ for differing frequency $f_p$ of the applied perturbation. The curves pertain to the last member $n = N$ in groups of size $N = 2$, 3, and 4. Member 2 displays a resonance peak near 0.6 Hz, a value associated with the effective spring constant or slope of the curves in Fig. 3c and the effective mass of the foil derived from its moment of inertia (see "Methods"). Member 3 displays amplified oscillations, and member 4 yet more so, oscillating wildly near resonance and resulting in collisions. This self-amplification property is the hallmark of flonons that we interpret as the root cause of the fragility of crystalline flocks.

## Wake interaction model and flock dynamical simulations

To understand the phenomena observed in experiments, we propose a minimal model based on an idealized treatment of the collective interactions. Motivated by the success of related formulations[28,33], we represent each flyer as a point particle that moves along a line while emitting a wake flow and interacting with the wake flows emitted by others. More abstractly, each particle (akin to the body of a bird) carries an oscillator (wings with prescribed flapping) that dictates the particle's self-propulsion as well as the signal (wake flow) left behind in its trail, as illustrated in the top panel of Fig. 4a. The particle experiences a propulsive force that depends on the interference between its instantaneous oscillator signal and the ambient wake signal left by others (middle panel of Fig. 4a). This general framework addresses, in a simplified but tractable way, the salient property that flocking involves interactions through long-lived flows that have memory of the earlier conditions under which they were generated[28].

The 1D nature of the ensemble and its motion, together with some reasonable assumptions about the form of the wake flow and the aerodynamic forces, permit a mathematical formulation. Briefly, the wake emitted at the location of flyer $n$ has vertical speed equal to its oscillator speed, $W_n = V_n$, and the flow speed thereafter decays

exponentially over time[28,29,33,34]. The next flyer $n+1$ experiences a thrust proportional to the square of its oscillator speed relative to the wake it encounters[33], i.e., $V_{n+1} - W_n$, and drag varies as the square of the flight speed $U_{n+1}$. These assumptions inform the complete derivations given as "Methods", yielding a system of differential equations with state-dependent delays and whose numerical solutions are discussed here.

The model equations are first used to determine the equilibrium gap spacing between flyers and their flight speed, these serving as initial conditions for simulations with $N = 5$ in a cyclic geometry like that of experiments (bottom panel of Fig. 4a). A representative example of the resulting dynamics is shown in Fig. 4b, and these data are animated in Supplementary Video 5. Like the robotic system, the simulated flock displays crystalline-like ordering into an array with a characteristic lattice spacing of about one wavelength. Also like the experiments, the ensemble displays "flonons" or displacement waves that are amplified during transmission down the group. The ensemble is also found to be fragile: Collisions result from members being initialized somewhat away from their equilibrium positions and speeds.

Steady and oscillatory perturbations applied to any member are implemented by adding appropriate forcing terms to the model equations (see "Methods"). The former case with $N = 4$ results in the force-displacement curves shown in the main plot of Fig. 4c. The high degree of overlap among the three data sets reflects the strictly one-way, nearest-neighbor interactions of the underlying model. The data obtained from simulations (dots) closely match the curve (dashed) corresponding to equilibrium solutions derived from the model equations. Comparison to Fig. 3c indicates that the force response curves have the same basic features as those measured in experiments, with quantitative differences in the equilibrium value of $S$ attributable to the model assumptions about flyer size and wake generation and

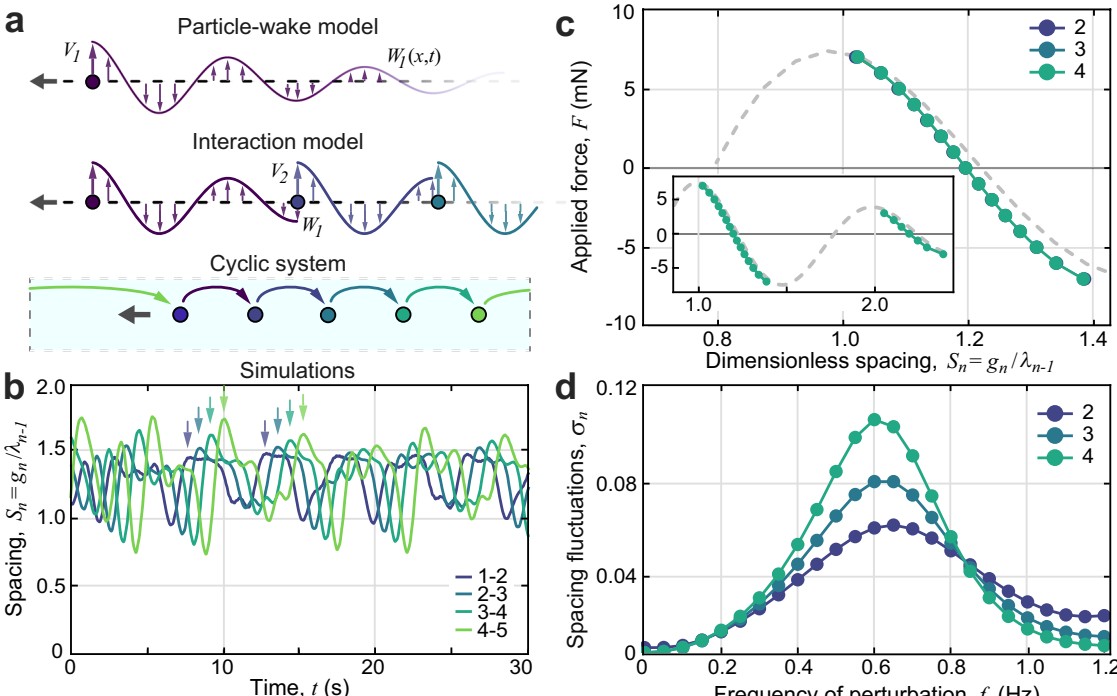

**Fig. 4 | A model of nonreciprocal interactions with memory and simulated flock dynamics. a** A model of wake generation and interaction. Each flyer emits a wake whose speed directly reflects its flapping speed, the flow thereafter decaying in time. The next flyer experiences thrust that depends on its flapping speed relative to the ambient wake. Cyclic boundary conditions mimic the closed system used in experiments. **b** Simulations of $N = 5$ members reveal crystal-like formations and

self-amplifying flonon modes (arrows). The conditions are chosen to match those of experiments. **c** Forces of interaction revealed by steady applied forces to downstream members for $N = 4$. **d** Resonant oscillations in downstream members in response to oscillatory forcing of the leader. The curves pertain to the last member in groups of size 2, 3, and 4.

interaction[29]. Further, the simulations confirm the existence of multiple stable positions, which occur near integer multiples of $S$, as reported previously for foil pairs[29]. The inset of Fig. 4c shows the force response associated with the primary and secondary wake positions ($S \approx 1$ and 2, respectively), the latter associated with a weaker slope or spring constant due to wake decay. Finally, oscillatory perturbations applied to the leader induce resonant amplification in later members, as characterized in Fig. 4d and animated in Supplementary Video 6. Collectively, these results indicate that the model, while a significant idealization of the hydrodynamics, nonetheless contains the essential physics needed to reproduce the salient phenomena.

Some subtle features from experiments are also replicated in simulations. For example, careful inspection of the experimental data in Fig. 3d reveals a slight shift to lower values of the resonant frequency for downstream members. The same trend, though perhaps of lesser degree, is seen in the simulation results of Fig. 4d. A contributing cause may be that later members undergo larger excursions from the equilibrium position, which, together with the nonlinear form of the $F(S)$ curves in Figs. 3c and 4c, leads to an effectively weaker spring constant[51].

The simulations allow for groups much larger than those studied in experiments and which give a sense of the behavior and trends for large $N$. Supplementary Video 7 shows a case with $N = 30$ in an open domain and with an AC perturbation applied to the leader serving as a simple model for disturbances encountered by the flock. The members tend to order into a lattice with approximately equal mean spacing, but later members oscillate increasingly strongly and eventually collide, at which point the simulation terminates. Hence, the stabilizing influence that promotes orderly positioning is upset by the destabilizing effect of the self-amplifying waves.

### Flonons are resonance cascades in crystals mediated by flyer-wake phasing

The model provides explanatory mechanisms for the existence of equilibrium positions, their local stabilization via restorative forces in pairwise interactions, and the global instability that results from many-body effects. The second panel of Fig. 4a helps to explain the pairwise effects. At the moment displayed, when the wings are in upstroke, the equilibrium position for flyer 2 is near the wake node that sits just upstream. If the flyer falls back as illustrated, it experiences an ambient wake flow that is generally counter to its flapping motion, leading to greater relative flow speed $V_2 - W_1$ and thus greater thrust that tends to restore the equilibrium. Flyer 3 shows the case of being ahead of the equilibrium node and the resulting decrease in relative speed and drop in thrust that is again restorative. Hence, the stabilization results from forces that vary with the phase of the flapping motions relative to the undulatory wake flow.

Masses stabilized by spring-like forces evokes the standard mass-spring array system[49], but the collective dynamics described here are fundamentally different from conventional normal modes since the interactions are one-way. To understand the consequences of such nonreciprocity, it is helpful to consider an idealized system consisting of a linear chain of masses linked to nearest neighbors by linear springs that have the special property of being directed in the leader-to-follower sense. Consider the response to a sinusoidal perturbation applied to the first member, whose dynamics are then completely determined because the one-way interaction does not permit member 2 (nor any other member) to influence member 1. For sufficiently weak damping and perturbation frequencies near or below resonance, the second member oscillates with greater amplitude than the first according to the classical results for a driven, damped harmonic oscillator[52]. This analysis exploits the fact that the 1–2 pair can be assessed as a system isolated from the rest of the flock, because the 2-to-3 link is one-way and hence the dynamics of 3 does not influence 2. With the dynamics of 2 completely determined, next proceed to the

2-3 subsystem, which again can be analyzed as isolated from all other downstream members. The same argument applies, and 3 resonates even more strongly than 2. Iterating pairwise and sequentially down the flock explains why the oscillations amplify. A complete mathematical analysis of this system gives the following result: The amplitude ratio between successive members is the standard gain factor for the driven, damped harmonic oscillator, where member $n$ plays the role of the driving source that forces member $n + 1$ to oscillate.

This analysis of a simplified but related system interprets flonons as resonance cascades driven by the downstream rectification and unstable amplification of vibrational energy. Similar phenomena have recently been reported in a mechanical system involving oscillations driven by motors whose nonreciprocal couplings are achieved electronically[53]. The degree of nonreciprocity was shown to control the signal transmission, notably leading to unidirectional amplification for highly asymmetrical interactions. Our findings show that such interactions are intrinsic to flow-mediated collective locomotion, suggesting that the resulting destabilizing effects may arise generically in natural flocks and schools.

### Diversity across individuals promotes stability of large, disordered flocks

The appearance of resonantly amplifying waves in experiments and simulations raises the issue of how such instabilities might be quelled so that longer and longer-lived formations can be realized. A cartoon view of a flight chain as masses linked by one-way springs suggests that the unstable growth of oscillations would be suppressed if members have sufficiently different resonant frequencies. This may be achieved with either different masses or different spring constants across individuals. To experimentally test the latter, we take advantage of the multiplicity of stable positions seen in follower-wake interactions (inset of Fig. 4c) and introduce a vacancy defect into the crystal. As illustrated in Fig. 5a, identical flyers each in the primary position of $S \approx 1$ downstream of the neighbor are expected to have identical spring constants, whereas a flyer downstream of a vacancy has $S \approx 2$ and a weaker spring due to wake decay. Supplementary Video 8 shows the experimental realization. As shown in the upper panel of Fig. 5b for a vacancy introduced between members 3 and 4 in a group of 5, measurements of the spacing fluctuations confirm that the last member has significantly smaller amplitude of its oscillations for the flock with a vacancy (magenta) than without (blue). Essentially, the oscillations are filtered out due to resonance frequencies that are incommensurate among members in the chain[52].

Unexpectedly, the muting of the instability does not show up in the motions of the member directly downstream of the vacancy, here member 4, who oscillates similarly strongly with and without the defect. This effect may seem to be at odds with the nearest-neighbor property, but in fact the same result is confirmed in our simulations, which have strictly nearest-neighbor wake interactions by construction of the underlying model. The fluctuation data from simulations are shown in the lower panel of Fig. 5b. The observed "skipping" of the member directly downstream of the vacancy seems to be a curious result of how fluctuations are transmitted around the group within the periodic system. Its appearance in simulations is a testament to the fidelity of the model and its ability to account for such unexpected outcomes. This odd feature notwithstanding, the larger point that vacancies mitigate the growth of the oscillations is well demonstrated in experiments and validated in simulations.

A more general strategy for suppressing flonon amplification involves introducing kinematic variability among the members. Within the model, this can be done, for example, by relaxing the assumption of uniform temporal phases in the flapping motions. Previous work on foil pairs has shown that changing the relative phase leads to changes in the equilibrium gap spacing that preserve the harmonious follower-wake phasing[33]. For example, a member that is out of phase, rather

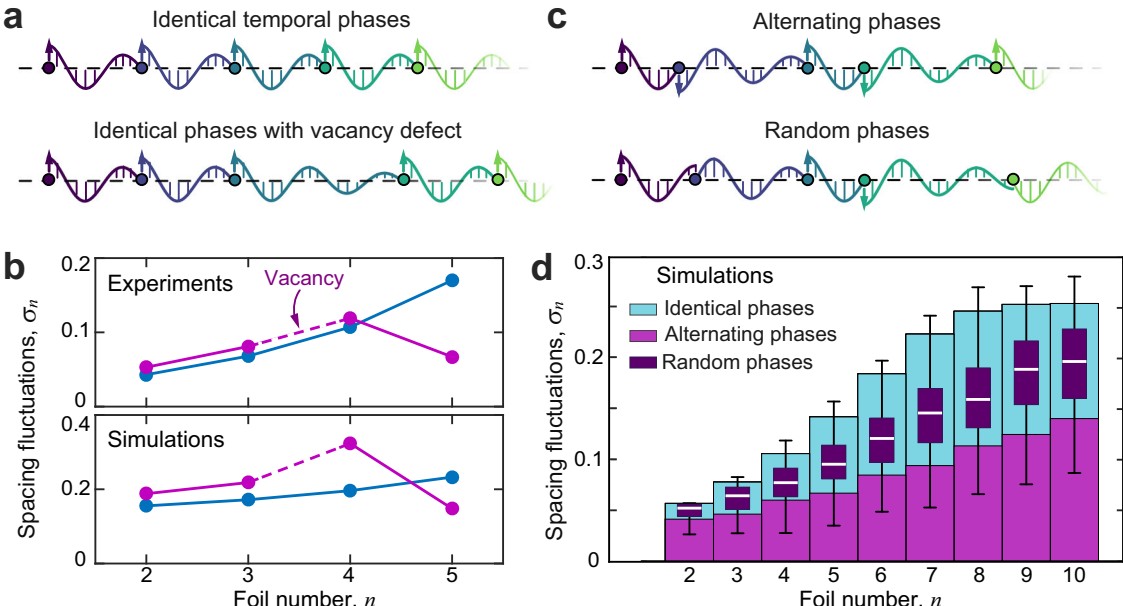

**Fig. 5 | Diversity across individuals promotes stability of the group. a** A member located at the $S \approx 2$ wake position sits downstream of a vacancy defect in the crystal. **b** Experiments and simulations on five-foil groups, with (magenta) and without (blue) a defect. The defect suppresses the fluctuations of the last member. **c** Varying the temporal phase of flapping across individuals leads to new equilibrium formations with different spacings. **d** Fluctuations in simulated flocks with $N = 10$ and different flapping phases. Identically in-phase flyers (cyan) experience larger fluctuations than those with alternating (magenta) and random (purple) phase relationships.

than in phase, with its upstream neighbor tends to sit near different wake nodes that roughly correspond to half-integer rather than integer values of $S$, as illustrated in the top panel of Fig. 5c. Because the phase controls the following distance, and because the interaction strength depends on the following distance due to wake decay, the phase also controls the spring constant for each member. In Fig. 5d, we compare simulations of flocks with $N = 10$ members that satisfy three different phase relationships: identically in-phase (cyan), out-of-phase with alternating vacancies (magenta) and random phases (purple) drawn from a uniform distribution such that $S \in [0.5, 2]$. These choices ensure comparable number density, and the flocks are initialized at their respective equilibrium configurations. A small-amplitude oscillatory force perturbs the leader throughout the run time, and the resulting fluctuations in spacing for other members are reported in Fig. 5d. The simulation results for the in-phase and out-of-phase cases are animated in Supplementary Videos 9 and 10. Compared to the identical-phase flock, groups with random phases tend to have lower fluctuations, as shown by the bar-and-whisker data that summarizes the results of many trials. The fluctuations are suppressed yet further in the alternating-phase group.

These results indicate that diversity across individuals can lead to formations with non-uniform spacing whose flonons are mitigated, providing a natural means for facilitating long flight chains. This effect ought to be enhanced by superposing different forms of individuality, as might be achieved within the model by varying other flapping kinematic parameters, body mass, and wing area. Presumably, bird flocks naturally possess such kinematic and morphological variability, which together with active sensing and behavioral response could allow for the very long formations shown in Fig. 1.

## Discussion

These results show that flocks of flapping flyers display novel structural and dynamical properties rooted in high-Reynolds-number flow interactions. Crystalline states in which members are ordered into lattices prove to be equilibrium configurations of identical individuals during tandem or in-line flight[27,29,31]. At the level of nearest-neighbor interactions mediated by flows, the lattice positions are stabilized by restoring forces that come about from proper phasing of the follower's undulatory motion with the leader's wave-like wake flow. However, longer chains of flyers are found to be fragile due to "flonons", a novel form of longitudinal displacement waves that grow in amplitude as they travel down the group and tend to cause collisions. These modes are qualitatively reminiscent of vibrational modes or phonons in conventional atomic and molecular crystals but distinctive in their amplification, a property that arises from the one-way flow interactions and which destabilizes large groups. Such interplay between cohesive forces and nonreciprocal amplification of fluctuations echoes and adds to recent studies in active matter[54,55]. We further show that long chains can be stabilized by structural disorder, introduced either directly through vacancy defects in the crystal or indirectly through variability or diversity across individuals. The latter arises naturally when members differ in the temporal phases of their flapping oscillations, which causes the group to lose conventional crystalline ordering of positions but retain harmonious phasing at the flyer-wake level.

Some of these phenomena may be exhibited by other forms of macroscale collective locomotion, since they can be traced back to a few common ingredients. First, the interactions are nonreciprocal, which ultimately derives from the inherent asymmetries of flows in the visco-inertial regime of intermediate to high Reynolds numbers[56]. Second, the interactions display memory or history dependence, which also stems from the inertial character of flows that causes their separation from bodies and the formation of wakes, vortices, jets, and other long-lived structures[25,28,56]. Third, preferred positioning is established by locomotors locking phase with spatially periodic patterns of such flows[27,29,30]. The first two elements seem to be generically present for locomotion at sufficiently high Re for which vortex wakes are produced, and future studies should investigate if a critical value must be exceeded to observe the phenomena of crystallization and flonons. The more nuanced effect of phase locking has recently been reported both in formation flight of birds and swimming fish. Specifically, a pair-wise analysis of in-line and echelon arrangements in migrating ibises revealed coherence in the wing tip paths[42]. Similarly,

the follower in echelon pairs of goldfish was shown to exhibit tail motions that lock onto the periodic vortices shed by the leader[57]. These encouraging correspondences motivate future work aimed at testing whether the phenomena revealed in our system are observed at the higher values of Re typical of flocking birds and schooling fish, as well as for more complex flapping kinematics involving pitching motions.

Our results may motivate further biological studies aimed at assessing the group-wide consequences of such interactions. While the systems investigated here are too idealized to expect quantitative correspondence, our findings present structural and dynamical phenomena to be assessed in animal groups and especially for linear formations of birds. They also inspire direct numerical simulations of the flow-structure interaction physics and behavioral models for actively controlling the observed collective instabilities. Reciprocally, biological systems will surely continue to motivate further investigations using robotic experiments[24]. For example, the prevalence of collective circling or milling across diverse animal groups[3,5,10,41,58–60] raises questions about whether such motions can emerge spontaneously from flow interactions. Whereas our system is constrained to orbital rotation, agents with additional dynamical degrees of freedom would be needed to address such problems.

Orderly formations among swimming and flying animals are typically ascribed to willful behavioral responses in which members actively take up and maintain positions that confer advantages for locomotion[10,13–15,41]. Our results show that formations can also arise in systems mediated only by physical interactions, much like crystal formation in conventional materials. Similarly, positional dynamics within animal groups might be attributed to purposeful maneuvers or perhaps variations in behavior[4,40,61], but our findings open up possible interpretations as intrinsic instabilities or other dynamical modes.

## Methods
### Experimental apparatus, procedures, and data analysis
Our experiments involve the rotational setup system shown in Fig. 2c, which extends previous flapping-foil systems[28,29,33,43,44] to accommodate many bodies, here up to five. The foils fan out from a common vertical shaft and, when flapped, propel in orbits around a water tank. Driven by a controlled stepper motor (not shown) mounted above the tank, the shaft oscillates up and down sinusoidally with prescribed amplitude $A$ and frequency $f$, imparting this flapping motion to all foils. Hence, the vertical position $A\sin(2\pi f t)$ and velocity $V = 2\pi A f \cos(2\pi f t)$ are identical for all foils. The data reported in this work correspond to $A = 1.5$ cm and $f = 2.5$ Hz, which are intermediate values in the accessible range. Each foil mounts to the shaft via an independent pair of rotary bearings, permitting free rotation under the fluid forces on the foil. In Fig. 2c, the housings that enclose the bearings are shown as stacked cylinders at the lower end of the shaft. A support arm connects each foil to its bearing housing, and all foils are held at a common radial distance $R = 19$ cm (measured at midspan) from the axis of rotation. The arms are bent up or down so that all foils lie within a common horizontal plane. The wings are thin airfoils 3D-printed from PLA plastic with NACA0017 cross-sectional profiles with chord length $c = 4$ cm. The planform shape is rectangular of span length $s = 8$ cm and thus area $sc = 32$ cm². 

Previous studies on foil pairs indicate that the interactions are dominated by hydrodynamics rather than mechanical coupling through the vertical shaft. Strong correspondence was found between systems with and without mechanical coupling. The former involved experiments on two foils connected to a single, common shaft via independent sets of rotary bearings[29], and the latter eliminated mechanical coupling through the use of two concentric shafts that do not come into contact and are driven by separate motors[33].

The clear-walled tank is rectangular with horizontal dimensions measuring 60 cm and height 30 cm, and a cylindrical inner wall of diameter 60 cm and height 30 cm provides an isotropic environment. A clear plastic lid eliminates waves and allows for undistorted imaging through the top, which is accomplished with a high-speed camera aimed downward at an angle. The camera captures the support arms, whose angles are tracked in the images in order to extract the angular positions of all foils. This is accomplished with a custom MATLAB program and a calibration procedure that involves imaging a polar grid for the conversion of apparent angle as seen by the camera to actual angle around the tank. The measured angles $\theta_n(t)$ of the foils are then converted to arc length or circumferential distance $X_n(t) = R\theta_n(t)$ around the tank. These data allow for the computation of flight speed $U_n = dX_n/dt$, trajectory wavelength $\lambda_n = U_n/f$, gap distance with the upstream neighbor $g_n = X_{n-1} - X_n - c$, and dimensionless spacing $S_n = g_n/\lambda_{n-1}$, all of which vary in time (see Fig. 2).

The chosen dimensions and experimental parameter values dictate dimensionless numbers that characterize the fluid dynamical regime. The Reynolds numbers are $Re_f = \rho A f c/\mu = 1500$ based on the typical flapping speed $Af$ and $Re = \rho U c/\mu = 1200$ based on the mean flight speed $U$ of a solo foil[43,44,56]. Here, $\rho$ is the density of water and $\mu$ its viscosity. The value of $Re \sim 10^3$ in our experiments is at the lower end of the range $Re \in [10^3, 10^5]$ relevant to schooling fish and flocking birds. Our measurements show that the flight speeds and therefore Re are modified on the order of 10% when in a group. Other dimensionless numbers include $A/c = 0.375$ and the Strouhal number $St = Af/U = Re_f/Re = 0.125$. These values, while not intended to match any particular biological system, fall within the broad ranges of values relevant to flapping locomotion of animals[11,25,47].

The application of external torques, both steady or DC and oscillatory or AC, is enabled by securing a targeted foil to the vertical shaft via a set screw in its bearing housing. Figure 3a illustrates the case of a DC perturbation accomplished via the scheme introduced by Ramananarivo et al.[29]. A steady force is to be applied to foil 2, whose underwater bearing housing is colored blue. The foil still propels freely thanks to an additional set of bearings that connect the upper portion of the shaft to the frame (white structure above the tank in Fig. 3a) that is oscillated by the motor. Hence the foil spins the shaft as it flies, and an external torque applied to the shaft is imparted to the foil. A steady perturbation is implemented with a weight attached to a string that is hung over a pulley and wound around a spool at the top of the shaft, where the sense of winding sets the direction of the applied torque. If sufficiently small, the external load causes the foil to take up a new position relative to its neighbors, where it resumes steady flight as the applied torque comes into balance with a hydrodynamic torque that reflects the flow interactions. We convert to effective force $F = Wr/R$ at the foil midspan, where $W$ is the applied weight and $r$ is the radius of the spool. The force-displacement curve is determined from repeated measurements of the modified position for differing magnitudes of the load applied in both directions. In this way, the forces of interaction are mapped out, yielding the results of Fig. 3c. Excessively large torques in either direction cause the foil to collide with a neighbor.

As shown in Fig. 3b, AC perturbations are implemented by locking the leader to the shaft, whose top end is now fitted with an on-board battery and small motor that drive back-and-forth oscillations of a small mass held out on an arm. Inertial coupling drives small oscillations of the shaft and thus the leader during its flight. The resulting spacing fluctuations for the downstream members are measured and plotted in Fig. 3d.

### Flapping locomotion in rotational and translational geometries
Rotational "flight mill" systems of the type used in these experiments have been validated in previous studies and shown to inform on translational locomotion at the level of qualitative phenomena and data trends. In some cases, but not generally, results from such systems show quantitative agreement. These claims are supported by a brief review of some key results found in rotational experiments

and corroborated by experiments, simulations and models of recti-linear propulsion: (1) A symmetric, rigid foil flapped at sufficiently high $Re_f$ spontaneously locomotes in a flight mill system[43,44], and this symmetry breaking bifurcation is also seen in 2D direct numerical simulations[45]. (2) Locomotion of a single foil in a flight mill is asso-ciated with the reverse von Karman wake of alternating vortices and jets[28,29,43,44], as seen in many studies of translational propulsion in physical and biological systems[25,47,62–64]. (3) Arrays of rigidly linked foils in a rotational setup display hysteretic propulsion dynamics[28,46], and the same is shown in 2D direct numerical simulations of a foil translating in a periodic domain[28]. The cause in both cases is coher-ent interactions of each flapper with the up-and-down jets between wake vortices. Similar phenomena are seen in systems with pure heaving[28] and combined heaving-pitching motions[46], indicating robustness to the form of flapping. (4) The system above maps onto an array of infinitely many foils freely propelling while held at pre-scribed spacing, and the results of rotational experiments match quantitatively well with a model of the translation dynamics of an infinite lattice emitting and interacting through point vortices[28,34]. (5) Independently locomoting and interacting foils in a pair sponta-neously order into formations in which the follower assumes one of several discrete locations behind the leader[29], and the same is seen in direct numerical simulations, vortex sheet simulations and point vortex models of translational propulsion[27,29,30]. The underlying mechanism is common across all systems. (6) Rotational experi-ments on independent, interacting foils in a pair reveal several dynamical modes when the kinematics of the follower are varied[33], and these results are reproduced with good quantitative agreement by a translational model based on thrust and drag[33]. Collectively, and along with the experimental and modeling results presented here, such correspondences indicate that rotational systems usefully inform on flapping locomotion phenomena.

Rotational systems differ from translational geometries due to self-interactions, i.e., the flows generated by the last member of a group may influence the leader. Such effects can be significant for strong driving kinematics and multiple bodies, both of which decrease the time elapsed between the generation of flows by the rear foil and when they are encountered by the leader. These inter-actions were characterized for two wings arranged diametrically opposite from one another via a common rigid support arm[28], in which case there is no distinct leader and follower. These experi-ments and associated modeling[34] indicated a flow decay timescale on the order of a second. Further experiments and modeling refined the estimate to $\tau \approx 0.5$ s[33], a value used in the model described below. For the five-wing system studied here, the strength of self-interactions can be estimated by comparing the effect of member 1 on 2 with that of 5 on 1. The conditions are such the typical time for a given foil to transit around the tank is $T \approx 4$ s (see Fig. 2d) and thus $T/\tau \approx 8$. Since the 5 foils are spread evenly around about half of the circumference, the inter-foil transit times are $T/8$ for 1-to-2 and $T/2$ for 5-to-1. Hence the effect of 5-to-1 interactions relative to 1-to-2 is approxi-mately $e^{-T/2\tau}/e^{-T/8\tau} = e^{-3T/8\tau} \approx e^{-3} \approx 5\%$. The small value indicates that the closed or periodic setting in the rotational experiments can usefully inform on open systems. (The introduction of a vacancy defect increases the effect to about $e^{-2} \approx 14\%$, which may explain the ele-vated fluctuations for member 1 seen in Fig. 5b.) The 5-to-1 interac-tion, while small in magnitude, is a persistent source of disturbances for the leader that likely contributes to the excitation of the flonons reported in Fig. 2. This claim is corroborated by simulations, which show similar fluctuations in a closed geometry (Fig. 4b), in contrast to the perfect crystalline ordering seen in open systems in the absence of any external perturbations. In essence, any perturbation to the leader induces amplified oscillations for downstream members, and self-interactions are an intrinsic source of such perturbations in the experiments.

## Formulation of mathematical model

The model formulation extends that introduced in Newbolt et al.[33]. Flyers are indexed by $n \in [1, N]$ and have instantaneous positions $X_n(t)$ and speeds $U_n(t) = \dot{X}_n(t)$ along the horizontal axis. Each carries an oscillator with prescribed signal $V_n(t) = 2\pi f_n A_n \cos(2\pi f_n t + \phi_n)$, which corresponds to the vertical speed of a wing flapping with frequency $f_n$, amplitude $A_n$, and temporal phase $\phi_n$. The flow speed $W_n(x, t)$ of the wake emitted by flyer $n$ is a function of space and time whose value at the flyer's location is assumed to be exactly the flapping speed, i.e., $W_n(X_n(t), t) = V_n(t)$, and this signal thereafter decays exponentially in time with timescale $\tau$ as a simplified treatment of wake dissipation. Hence, each flyer 'writes' an oscillatory signal into the fluid in its trail, as shown schematically in the top panel of Fig. 4a.

Each flyer is assumed to be an inertial body subject to thrust and drag. In the absence of interactions, these aerodynamic forces take the conventional forms $\rho C_T s c V_n^2/2$ and $\rho C_D s c U_n^2/2$, with thrust varying quadratically with the flapping (vertical) speed and drag with the propulsion (horizontal) speed. Here $\rho$ is the density of fluid, $sc$ is the planform area (span times chord) of the propulsor (wing or foil), and the coefficients of thrust $C_T$ and drag $C_D$ are dimensionless quantities. Aerodynamic interactions are captured by a relative flow model in which the thrust is modified to depend on the vertical speed of the oscillator relative to any ambient wake signal. Additionally, we assume an 'erase-and-replace' scheme in which each flyer overwrites the wake signal of its upstream neighbor with its own signal, as illu-strated in the middle panel of Fig. 4a, in which case the relative flow is $V_n(t) - W_{n-1}(X_n(t), t)$. Hence, the interactions are nearest-neighbor and one-way or nonreciprocal: Each follower in a pair is forced by, but cannot force, the leader.

The propulsion dynamics for each flyer of mass $M$ (assumed identical for all $n$) is dictated by Newton's Second Law:

$$M\dot{U}_n(t) = \frac{1}{2}\rho C_T s c \left[ V_n(t) - V_{n-1}(t_n(t))e^{-(t-t_n(t))/\tau} \right]^2 - \frac{1}{2}\rho C_D s c U_n^2(t).$$

(1)

The interaction term in square brackets is nontrivial since it involves the earlier time $t_n(t)$ when the upstream neighbor $n-1$ was at the current location of flyer $n$, as defined through the implicit relationship $X_n(t) = X_{n-1}(t_n(t))$. Hence, $t_n(t)$ embodies the effect of memory in the system. Equivalently, $t - t_n$ is the relevant delay time between when a signal left by flyer $n-1$ is encountered by flyer $n$. The memory time-scale is itself time dependent, motivating an approach in which it is included as a state variable in the dynamical system by deriving its evolution equation: $\dot{X}_n(t) = \dot{X}_{n-1}(t_n(t))\dot{t}_n(t)$ and thus $\dot{t}_n(t) = \dot{X}_n(t)/\dot{X}_{n-1}(t_n) = U_n(t)/U_{n-1}(t_n)$. This leads to a system of non-linear delay differential equations for the state variables $(X_n, U_n, t_n)$:

$$\dot{X}_n(t) = U_n(t)$$
$$\dot{U}_n(t) = \frac{\rho C_T s c}{2M}\left[V_n(t) - V_{n-1}(t_n(t))e^{-(t-t_n(t))/\tau}\right]^2 - \frac{\rho C_D s c}{2M}U_n^2(t)$$

(2)

$$\dot{t}_n(t) = U_n(t)/U_{n-1}(t_n(t))$$

Here the term $U_{n-1}(t_n(t))$ in the third equation constitutes a state-dependent delay[65], and all time dependencies are explicitly included for clarity. A flock of $N$ flyers is governed by $3N$ such equations.

The treatment of the leader $n = 1$ depends on whether the system is open or closed, i.e., whether the ensemble flies into a semi-infinite domain of quiescent fluid (like a flock of birds) or within a cyclic domain (as in our experiments). If open, then the leader flies exactly as a solo bird: The interaction term involving $V_{n-1}$ in the second equation of the system [(2)] is removed, as is the entire third equation. If closed, then the leader interacts with the last member of the flock, and $n-1$ should be replaced by $N$ in the second and third equations of the system [(2)]. The latter case is illustrated in the bottom panel of Fig. 4a.

## Model parameters and numerical solutions

The constants appearing in the model are informed by the corresponding experimental quantities. Considering the rotational-translational analogues of angle with distance and torque with force[33], the effective mass $M = I/R^2 = 20$ g in the model is related to the experimental parameters of the moment of inertia $I$ (of the foil, support arm, and bearing housing) and arm radius $R$. The propulsor area $sc = 32$ cm$^2$ is taken from experiments, and $\rho = 1.0$ g/cm$^3$ for water. In experiments, the circumferential length around the tank is $2\pi R = 119$ cm, of which the fluid occupies a reduced length $2\pi R - 5c = 99$ cm due to the presence of the 5 foils. The simulation runs reported here employ a value of $L = 91$ cm, which was determined by trial and error to produce schooling number fluctuations qualitatively similar to experiments. The thrust $C_T = 0.96$ and drag $C_D = 0.074$ coefficients are taken from previous studies that used the same foil shape[33], as is the wake decay timescale $\tau = 0.5$ s. The values $A_n = A = 1.5$ cm and $f_n = f = 2.5$ Hz match those used in experiments. The temporal phases $\phi_n$ are varied for different simulations, as discussed in more detail below.

All model results are furnished by numerically integrating the governing equations using MATLAB's solver *ddesd* for delay differential equations with state-dependent delays[65]. An adaptive time step scheme ensures relative errors of no more than $10^{-3}$. All simulation runs reported here are initialized at the equilibrium configuration and speeds, as determined by numerically analyzing the model equations. The results of Fig. 4 involve $\phi_n = 0$, and Fig. 4b displays the unperturbed dynamics for a group of $N = 5$ in a closed system. In all plots of the simulation outputs, the gap distance is defined as $g_n = X_{n-1} - X_n$, a form appropriate to the point-like particles ($c = 0$) in the model. The dimensionless spacing is $S_n = g_n/\lambda_{n-1}$ with values of $\lambda_n$ approximated by their equilibrium values.

The application of DC or steady forces to a flyer $n$ is accomplished by adding a constant term $F/M$ on the right hand side of the $\dot{U}_n$ equation in the system [(2)]. Each data point of Fig. 4c is obtained for $N = 4$ in a closed system. Each curve corresponds to perturbations applied to $n = 2, 3$ and $4$, respectively, for which $F$ is incrementally varied to determine the resulting mean spacing $S$ achieved at sufficiently long times.

The application of AC or oscillatory perturbations to the leader is accomplished by adding a term $(F_{AC}/M)\sin(2\pi f_p t)$ to the right hand side of the $\dot{U}_1$ equation in the system [(2)]. The forcing amplitude $F_{AC} = Kf_p^2$ is frequency dependent with a form that is derived from an analysis of the experimental perturbation device. The oscillating motor enforces a relative angle $\theta_p \sin(2\pi f_p t)$ between the foil (and its supporting structures) and the weighted arm of the coupled oscillator. Assuming the two are coupled only through inertia, disregarding drag on the foil, and demanding that angular momentum is conserved leads to an angular acceleration of the foil that varies sinusoidally with an amplitude of $\theta_p(2\pi f_p)^2/(1 + I/I_p)$, where $I_p$ is the moment of inertia of the weighted arm. The usual rotation-to-translation conversions then lead to the forcing form given above with the constant prefactor $K = 4\pi^2 MR\theta_p/(1 + I/I_p) = 442$ g·cm, whose value is estimated from experimental parameters.

Each curve of Fig. 4d represents the last member $n = N$ in groups of size $N = 2, 3$, and $4$. The system is open, and AC perturbations are applied to the leader with $f_p$ varied up to 1.2 Hz. This procedure explores force amplitudes $F_{AC}$ up to about 6 mN = 600 dyn. The data of Fig. 5d correspond to $N = 10$ and an open system, and the leader $n = 1$ is perturbed with an oscillatory force term with $F_{AC} = 1.6$ mN = 160 dyn and $f_p = 0.6$ Hz, which is the resonant frequency for small displacements about the equilibrium position. The various relationships for the temporal phases $\phi_n$ are specified in the text and caption.

## Data availability

The data generated for this study are plotted in the manuscript figures, and all source data are provided in a supplementary file. Source data are provided with this paper.

## Code availability

All relevant codes are available upon request.

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

## Acknowledgements

We are grateful to P. Chaikin, S. Childress, E. Kanso, B. Kohn, A. Oza, C. Peskin, M. Shelley, and J. Zhang for useful discussions. This work was supported by the U.S. National Science Foundation through grants DMS-1847955 and DMS-1646339 to L.R.

## Author contributions

J.W.N., S.R., and L.R. conceived of the study; J.W.N., N.L., M.B., and L.R. conducted the experiments and analyzed the data; J.W.N., J.W., C.M., and L.R. formulated the model and carried out the simulations; and all authors contributed to interpreting the results and composing the manuscript.

## Competing interests

The authors declare no competing interests.
