## [Peer Review File · Nature Communications]

REVIEWER COMMENTS

Reviewer #1 (Remarks to the Author):

The authors report on the structure and dynamical stability of groups of self-propelling foils in a water tank. Their paper is very well written, and the results make a significant and robust contribution to the literature on collective locomoting animals.

The experimental system resembles that of *Daphnia* (commonly called water fleas) which circulate around shafts of light that penetrate the water. Single *Daphnia* and *Daphnia* in dilute concentration circulate erratically and independently around the light shafts. *Daphnia* in high concentration circulate collectively in one direction, forming a vortex. The collective behaviour may be attributed to hydrodynamic coupling since the water inside the vortex turns in the same direction as the *Daphnia*.

See Erdmann et al. Active Brownian Particle and Random Walk Theories of the Motions of Zooplankton: Application to Experiments with Swarms of *Daphnia*. 2004 arXiv.

See also Erdmann, U., Ebeling, W., 2003. Collective motion of Brownian particles with hydrodynamic interactions. *Fluctuation Noise Lett.* 3, L145–L154.

This has resonance with the author's study.

The counter view is that the collective motion of the *Daphnia* is the result of repulsion between individuals.

See Ordemann, A., Balázsi, G., Caspari, E., & Moss, F. *Daphnia* swarms: from single agent dynamics to collective vortex formation. In *Fluctuations and Noise on Biological, and Biomedical Systems* (Ed. S.M. Bezrukov, S.M., Frauenfelder, H. & Moss, F. Proc. of SPIE Vol. 5110) (2003).

By way of contrast, localized swarms of airborne insects do not circulate collectively in one direction.

See, e.g., Okubo, A. Dynamical aspects of animal grouping: swarms, schools, flocks, and herds. *Adv. Biophys.* 22, 1-94 (1986).

Kelley, D.H. & Ouellette, N.T. Emergent dynamics of laboratory insect swarms. *Sci. Rep.* 3, 1073, 1-7 (2013).

These results raise two important issues that the authors need to address/discuss.

First, what is the role of Reynolds number? Hydrodynamic interactions between Daphnia may give rise to coherent collective behaviour whereas aerodynamic interactions between insects seem not too. Second, the quasi-linear flight formations of birds (particularly smaller birds that create complex wakes rather than long-winged birds) which motivated the current study may not be due to aerodynamic interactions but may instead be attributed to mutual repulsion. Reynolds numbers quantifying bird flights are typically between 10^3 to 10^5 and so typically much larger than the Reynolds number, ~ 1500 , attained in the experimental study. Note that Reynolds numbers quantifying the circulating Daphnia are typically between 1 and 10.

Consequently, statements about the potential relevancy of the water tank study to swarming insects and flocking birds (the motivation behind the experimental study) needs to be qualified. The issue is glossed over on page 9 where the authors write “These values [Reynolds number and Strouhal number], while not intended to match any particular biological system, fall within the broad ranges of values relevant to high-Re flapping locomotion of animals [7,21,52].” This is true of the Strouhal number [52] but the references to Reynolds numbers [7,12] pertain to small aquatic animals (e.g., larvae) rather than to animals in general.

Reviewer #2 (Remarks to the Author):

The manuscript reports on a primarily experimental study concerning the group stability of self-propelling flapping hydrofoils. The examined system is an adaptation of the author’s prior works and can be viewed as an abstraction of a finite-size bird flock. Using this setup, the authors show that the interaction between leaders and followers in their mock flock is similar to one-way springs, leading initially to lattice-like self-organization of the foils making up the flock. However, it is found that the same non-reciprocal flow-induced interaction is also responsible for the growth and propagation of positional waves that eventually upset the orderly formation of the flock. The authors propose a simple mathematical model that can replicate some of the observed behaviors in the experiments. Moreover, they demonstrate how the amplification of flow-induced oscillations can be avoided by purposefully introduced positional disorder stemming from variability in propulsion characteristics of the individuals in the flock.

Overall, the manuscript is well-written, the methodology is sound, and the results are valuable additions to the literature on collective flying/swimming. I am, therefore, in favor of publishing the manuscript in Nature Communications. My comments for improving the quality of the manuscript are:

- The authors may want to discuss whether they expect the same grouping dynamics for pitching and simultaneously pitching and heaving foils.

- In the third paragraph of page 3, the authors state: "These dynamics were not reported in recent many-flapper simulations [27] conducted at lower $Re = 200$, suggesting the primary importance of inertial effects." Does this mean there is a critical Reynolds number beyond which positional disturbances grow unstably? Can this be further explored by systematically repeating the experiments at lower amplitudes and frequencies? Moreover, if it exists, does this critical Reynolds number also depend on the number of foils in the array?

- In the first paragraph of page 7, the authors state: "For a perturbation frequency sufficiently near resonance, and for sufficiently weak damping, the second member oscillates with greater amplitude than the first according to the classical results for a driven, damped harmonic oscillator." It would be more informative if "sufficiently near resonance" and "sufficiently weak damping" are quantified in relative terms.

- The authors may wish to cite the recent review of Timm et al. (Multi-body hydrodynamic interactions in fish-like swimming, *Appl. Mech. Rev.*, 2024) in paragraph three of the introduction section, where they reference prior studies on multi-propulsor interactions.

Response to referee reports for “Flow interactions lead to self-organized flight formations disrupted by self-amplifying waves”
(NCOMMS-23-35574)

Response to Reviewer #1

We thank the referee for this encouraging assessment and for the many useful recommendations aimed at clarifying the implications of our study. Below we reproduce the report in full and give point-by-point responses that include summaries of the revisions made to the manuscript, where changes appear in red text. With these improvements, we hope the referee will reconsider the work.

The authors report on the structure and dynamical stability of groups of self-propelling foils in a water tank. Their paper is very well written, and the results make a significant and robust contribution to the literature on collective locomoting animals.

We thank the referee again for this synopsis and positive overall assessment.

The experimental system resembles that of *Daphnia* (commonly called water fleas) which circulate around shafts of light that penetrate the water. Single *Daphnia* and *Daphnia* in dilute concentration circulate erratically and independently around the light shafts. *Daphnia* in high concentration circulate collectively in one direction, forming a vortex. The collective behaviour may be attributed to hydrodynamic coupling since the water inside the vortex turns in the same direction as the *Daphnia*.

See Erdmann et al. Active Brownian Particle and Random Walk Theories of the Motions of Zooplankton: Application to Experiments with Swarms of *Daphnia*. 2004 arXiv.

See also Erdmann, U., Ebeling, W., 2003. Collective motion of Brownian particles with hydrodynamic interactions. *Fluctuation Noise Lett.* 3, L145–L154.

This has resonance with the author’s study.

The counter view is that the collective motion of the *Daphnia* is the result of repulsion between individuals.

See Ordemann, A., Balázsi, G., Caspari, E., & Moss, F. *Daphnia* swarms: from single agent dynamics to collective vortex formation. In *Fluctuations and Noise on Biological, and Biomedical Systems* (Ed. S.M. Bezrukov, S.M., Frauenfelder, H. & Moss, F. Proc. of SPIE Vol. 5110) (2003).

By way of contrast, localized swarms of airborne insects do not circulate collectively in one direction.

See, e.g., Okubo, A. Dynamical aspects of animal grouping: swarms, schools, flocks, and herds. *Adv. Biophys.* 22, 1-94 (1986).

Kelley, D.H. & Ouellette, N.T. Emergent dynamics of laboratory insect swarms. *Sci. Rep.* 3, 1073, 1-7 (2013).

We thank the referee for raising these related biolocomotion systems and bringing these interesting papers to our attention. The phenomena displayed by *Daphnia* are especially interesting and new to us. We appreciate the many parallels with our system, especially with regard to hydrodynamically interacting agents undergoing collective orbital motions. Correspondingly, we have added citations to these works and expanded our discussions in the introduction and conclusions sections. Please see pages 1 and 9 of the revised manuscript as well as the expanded bibliography. In particular, it is interesting that collective circling is seen in many animal systems, including birds (e.g. swirling flocks and murmurations), fish (milling form of schooling and so-called “bait balls”), and also of course *Daphnia* vortexing as raised by the referee. It would be fascinating to devise a robotic experiment capable of addressing whether such motions can emerge spontaneously from the flow interactions. This future direction would require a significant advance over our current system in which the orbital rotation comes about as a constraint on the motions of the flappers, as described in the Methods under subsection “Flapping locomotion in rotational and translational geometries.”

These results raise two important issues that the authors need to address/discuss.

First, what is the role of Reynolds number? Hydrodynamic interactions between *Daphnia* may give rise to coherent collective behaviour whereas aerodynamic interactions between insects seem not to.

We thank the referee for these thought provoking questions. While we cannot say with certainty, we believe the issue being raised is more about the form of locomotion rather than the Reynolds number *per se*. The Reynolds number ranges for swimming *Daphnia* and most flying insects differ somewhat but both are sufficiently inertial to leave wake flows with which others can in principle interact. The main difference we see is that, since *Daphnia* enjoy the buoyancy of living in water, their flows are likely associated with thrust or propulsion and hence their wakes trail behind them as they swim along. Hence, *Daphnia* chasing one another have a good chance to interact through their wakes. This is broadly similar to schooling interactions among fish, albeit different in the value of Re and mode of propulsion. In contrast, flying insects are dominantly hoverers concerned with lift generation for weight support and therefore the wakes from their flapping wings are primarily streaming downward. Correspondingly, we are not aware of any coherent collective movement among flying insects that can be attributed specifically to aerodynamic interactions. Birds differ from insects in that their lift generation mechanism relies on translation through air during flight, and it is well documented that the wakes, while deflecting downwards somewhat, are close enough to horizontal that trailing birds can interact. It is for these reasons that the discussions in our manuscript emphasize parallels to bird flight and fish swimming rather than insect flight. We have also removed the reference to insect swarms in the Abstract.

About the general role played by Reynolds number, some relevant discussions are contained in the paragraph spanning pages 8 and 9. The flow regime must be sufficiently inertial so that vortices

form and are shed into wakes that persist for long enough times that others may interact. More nuanced is the particular structure of the wake, which is well documented to be the reverse von Karman array of alternating vortices for flappers operating with bird- and fish-like parameters. This issue is discussed in some detail in the Methods subsection “Flapping locomotion in rotational and translational geometries.”

Second, the quasi-linear flight formations of birds (particularly smaller birds that create complex wakes rather than long-winged birds) which motivated the current study may not be due to aerodynamic interactions but may instead be attributed to mutual repulsion. Reynolds numbers quantifying bird flights are typically between 103 to 105 and so typically much smaller than the Reynolds number, ~ 1500 , attained in the experimental study. Note that Reynolds numbers quantifying the circulating *Daphnia* are typically between 1 and 10.

Consequently, statements about the potential relevancy of the water tank study to swarming insects and flocking birds (the motivation behind the experimental study) needs to be qualified. The issue is glossed over on page 9 where the authors write “These values [Reynolds number and Strouhal number], while not intended to match any particular biological system, fall within the broad ranges of values relevant to high-Re flapping locomotion of animals [7,21,52].” This is true of the Strouhal number [52] but the references to Reynolds numbers [7,12] pertain to small aquatic animals (e.g., larvae) rather than to animals in general.

We agree that the Reynolds number $Re \sim 10^3$ in our experiments is on the low end of the range $Re \in [10^3, 10^5]$ relevant to flocking birds and schooling fish. We have now clarified this both in the text of the paper (see page 2) when we first introduce the experimental system as well as in the Methods section (pages 9-10). Further, per the referee’s suggestion, we have qualified our statements about how our results relate to animal groups by revising the discussion on page 9. While the value of Re is somewhat low, we emphasize that our system parameters are such that the characteristic reverse von Karman wake is generated. This issue is discussed in the Methods subsection “Flapping locomotion in rotational and translational geometries.” The gross wake structure that dictates the flow interactions is therefore relevant to flapping-based propulsion of animals, although the details are surely different.

Response to Reviewer #2

We thank the referee for this encouraging assessment and for the helpful recommendations. Below we reproduce the report and address each comment with point-by-point responses that include summaries of the revisions made to the manuscript, where changes appear in red text. We are grateful for the referee’s guidance in improving the manuscript, which we hope will be reconsidered.

Overall, the manuscript is well-written, the methodology is sound, and the results are valuable additions to the literature on collective flying/swimming. I am, therefore, in favor of publishing the manuscript in Nature Communications.

We thank the referee again for this praise and positive evaluation.

The authors may want to discuss whether they expect the same grouping dynamics for pitching and simultaneously pitching and heaving foils.

We thank the referee for raising this interesting point. Indeed, previous experiments on rigidly linked but freely propelling foils provide good evidence that the flow interactions are of a similar nature for pure heaving and combined heaving-pitching motions. In particular, our previous experiments in ref. 28 with purely heaving foils compare favorably with results from ref. 46 on hinged foils that undergo pitching. Per the referee’s suggestion, we raise this issue of flapping kinematics in the discussion of the paper (page 9) as well as in the Methods (page 10).

In the third paragraph of page 3, the authors state: “These dynamics were not reported in recent many-flapper simulations [27] conducted at lower $Re = 200$, suggesting the primary importance of inertial effects.” Does this mean there is a critical Reynolds number beyond which positional disturbances grow unstably? Can this be further explored by systematically repeating the experiments at lower amplitudes and frequencies? Moreover, if it exists, does this critical Reynolds number also depend on the number of foils in the array?

We thank the referee for these interesting questions. We cannot be certain as to why the previous simulations (now ref. 31) did not observe these waves since this work is different from ours in many ways. It involves 2D simulations of lower- Re propulsion for an array of flexible filaments undergoing complex undulations as they move into quiescent fluid. We do believe that sufficiently high Re is needed to see unstable waves, as justified by the basic requirements that wakes form by vortex shedding and are sufficiently long-lived due to inertia. These issues are covered in the discussion spanning pages 8 and 9. But we also see that it is important that some ambient noise be present to trigger the waves. Disturbances to the leader are intrinsic to our experiments given the rotational geometry, and the same is true of our simulations with cyclic or periodic boundary conditions. Our simulations of the non-periodic case in which the flappers move into quiescent fluid require forcing perturbations to the leader to excite the waves. So, it may be that the previous study (ref. 31) did not see these waves since there was no source of disturbance. Accordingly, we have modified the associated text spanning pages 3 and 4 to include this possible interpretation.

How the behavior changes when Reynolds number is increased from low to moderate and high values would certainly be interesting to study in the future. We have revised the discussion to include this as a follow-up direction (page 9). Our current experimental system is unfortunately limited with regard to the range of Re that is accessible while maintaining reliability of the measurements. Too high f and A lead to fast propulsion that causes the leader to interact strongly with the last member’s wake, while too low values cause the propulsive dynamics to be contaminated by bearing friction. In practice, this means that f and A can be varied up or down by about factors of 2 relative the values used in our study. In casual explorations over this range, we have observed qualitatively the same behaviors as reported in the paper.

We think that the presence or absence of amplifying fluctuations is independent of the number of flyers. The underlying instability is essentially a two-body or nearest-neighbor effect that, when repeated pairwise down the whole group, shows up as an unstably growing traveling wave in the relative displacements. The detailed argument supporting this view is provided in the middle paragraph of page 7. Our experimental data also support this view, e.g. the AC perturbations characterized in Fig. 3d lead to amplification for 2-member groups as well as 3- and 4-flapper groups. In our simulations, the waves show up for groups of all sizes, with the main difference being that longer groups yield higher amplitudes, as displayed by the furthest downstream members (e.g. see Fig. 5d).

In the first paragraph of page 7, the authors state: “For a perturbation frequency sufficiently near resonance, and for sufficiently weak damping, the second member oscillates with greater amplitude than the first according to the classical results for a driven, damped harmonic oscillator.” It would be more informative if “sufficiently near resonance” and “sufficiently weak damping” are quantified in relative terms.

We can address the referee’s point by considering a damped spring-mass system that is driven by sinusoidal motion of its support. The mass plays the role the follower in a pair, and the support is the leader. Adopting standard notations, the equation of motion of the mass is $\ddot{x} + 2\zeta\omega_0\dot{x} + \omega_0^2(x - x_0) = 0$ that, with the driving $x_0(t) = A_0 \sin \omega t$, becomes $\ddot{x} + 2\zeta\omega_0\dot{x} + \omega_0^2x = \omega_0^2A_0 \sin \omega t$. Hence the system is equivalent to the classic damped harmonic oscillator with resonance frequency $\omega_0 = \sqrt{k/m}$, damping constant c and its dimensionless form $\zeta = c/2m\omega_0$, and subject to sinusoidal forcing of amplitude $\omega_0^2A_0$ and frequency ω . This system has the gain factor $G = A/A_0 = \omega_0^2/Z\omega$ with $Z = \sqrt{(2\omega_0\zeta)^2 + (\omega_0^2 - \omega^2)^2}/\omega^2$, which can be cast in dimensionless form as

$$G(\bar{\omega}, \zeta) = \frac{1}{\bar{\omega}\sqrt{(2\zeta)^2 + (1 - \bar{\omega}^2)^2/\bar{\omega}^2}} \quad \text{where} \quad \bar{\omega} = \frac{\omega}{\omega_0}.$$

Some resonance curves $G(\bar{\omega})$ for fixed ζ are shown in the figure below, along with the color map of $G(\bar{\omega}, \zeta)$. The latter shows the region of parameter space with $G > 1$ (red). These results quantify that amplification occurs for sufficiently low damping and for driving frequencies ω near to or less than the resonance frequency ω_0 . The $G = 1$ contour (dashed curve in the right panel) delineates the amplification zone. While these quantifications are difficult to succinctly describe, we have revised the text on page 7 to indicate that driving frequencies below resonance can also cause amplification.

Figure 1: Amplification in the classic driven, damped harmonic oscillator. *Left*: Resonance curves, as quantified by the gain versus dimensionless driving frequency $G(\omega/\omega_0)$, where each curve has fixed dimensionless damping constant ζ . *Right*: Map of the gain $G(\omega/\omega_0, \zeta)$, where the dashed curve is the $G = 1$ contour. Amplification ($G > 1$, red region) occurs for sufficiently low damping and low driving frequency.

The authors may wish to cite the recent review of Timm et al. (Multi-body hydrodynamic interactions in fish-like swimming, *Appl. Mech. Rev.*, 2024) in paragraph three of the introduction section, where they reference prior studies on multi-propulsor interactions.

We thank the referee for bringing this excellent and timely review to our attention, and we have now included the citation in the introduction.

REVIEWERS' COMMENTS

Reviewer #1 (Remarks to the Author):

You have done an excellent job in responding to my comments, and consequentially I recommend that your manuscript be published in Nature Communications.

Reviewer #2 (Remarks to the Author):

I am satisfied with the authors' responses and revisions, and, therefore, am happy to recommend the manuscript for publication in Nature Communications.